# Discrimination between the Two Closely Related Species of the Operational Group *B. amyloliquefaciens* Based on Whole-Cell Fatty Acid Profiling

**DOI:** 10.3390/microorganisms10020418

**Published:** 2022-02-11

**Authors:** Thu Huynh, Mónika Vörös, Orsolya Kedves, Adiyadolgor Turbat, György Sipos, Balázs Leitgeb, László Kredics, Csaba Vágvölgyi, András Szekeres

**Affiliations:** 1Department of Microbiology, Faculty of Science and Informatics, University of Szeged, Közép Fasor 52, H-6726 Szeged, Hungary; huynh_thu@hcmut.edu.vn (T.H.); voros.monesz@gmail.com (M.V.); kedvesorsolya91@gmail.com (O.K.); adiyadolgor_turbat@yahoo.com (A.T.); kredics@bio.u-szeged.hu (L.K.); mucor1959@gmail.com (C.V.); 2Department of Biotechnology, Faculty of Chemical Engineering, Ho Chi Minh University of Technology (HCMUT), 268 Ly Thuong Kiet Street, District 10, Ho Chi Minh City 72607, Vietnam; 3Vietnam National University Ho Chi Minh City, Linh Trung Ward, Thu Duc District, Ho Chi Minh City 71351, Vietnam; 4Functional Genomics and Bioinformatics Group, Research Center for Forestry and Wood Industry, University of Sopron, Bajcsy-Zsilinszky Str. 4, H-9400 Sopron, Hungary; sipos.gyorgy@uni-sopron.hu; 5Institute of Biophysics, Biological Research Centre, Eötvös Loránd Research Network, Temesvári Krt. 62, H-6726 Szeged, Hungary; leitgeb@brc.hu

**Keywords:** *Bacillus* taxonomy, *Bacillus velezensis*, *Bacillus amyloliquefaciens*, fatty acid profiling, chemotaxonomy

## Abstract

(1) Background: *Bacillus velezensis* and *Bacillus amyloliquefaciens* are closely related members of the “operational group *B. amyloliquefaciens*”, a taxonomical unit above species level within the ”*Bacillus subtilis* species complex”. They have similar morphological, physiological, biochemical, phenotypic, and phylogenetic characteristics. Thus, separating these two taxa from each another has proven to be difficult to implement and could not be pushed easily into the line of routine analyses. (2) Methods: The aim of this study was to determine whether whole FAME profiling could be used to distinguish between these two species, using both type strains and environmental isolates. Initially, the classification was determined by partial sequences of the *gyrA* and *rpoB* genes and the classified isolates and type strains were considered as samples to develop the identification method, based on FAME profiles. (3) Results: The dissimilarities in 16:0, 17:0 iso, and 17:0 FA components have drawn a distinction between the two species and minor differences in FA 14:0, 15:0 iso, and 16:0 iso were also visible. The statistical analysis of the FA profiles confirmed that the two taxa can be distinguished into two separate groups, where the isolates are identified without misreading. (4) Conclusions: Our study proposes that the developed easy and fast-automated identification tool based on cellular FA profiles can be routinely applied to distinguish *B. velezensis* and *B. amyloliquefaciens*.

## 1. Introduction

*Bacillus velezensis* and *B. amyloliquefaciens*, together with the *B. siamensis* and a black-pigment-producing strain, *B. nakamurai*, are members of the operational group *B. amyloliquefaciens.* This operational group belongs to the *B. subtilis* species complex with an eventful taxonomic history [1]. Both taxa are beneficial species, they have played increasingly important roles in applied microbiology [2,3,4,5]. Furthermore, several plant growth-promoting and biocontrol products from *B. amyloliquefaciens* and *B. velezensis* are now commercially available, including RhizoVital^®^ (*B. velezensis* DSM 23117^T^; ABiTEP, GmbH, Berlin, Germany), Amylo-X^®^ WG (*B. amyloliquefaciens* subsp. *plantarum* D747; Certis Europe BV, Utrecht, The Netherlands), RhizoPlus^®^ (*B. amyloliquefaciens* FZB24; ABiTEP), and Taegro^®^ (*B. subtilis* var. *amyloliquefaciens* FZB24; Novozymes Biologicals, Inc., Salem, VA, USA) [6].

*B. amyloliquefaciens* was first described by Fukumoto [7,8] and later revised by Priest et al. [7,8] as an industrial producer of amylase, while *B. velezensis*, first isolated from the river Vélez in Málaga (southern Spain), was initially described as a distinct ecotype of *B. amyloliquefaciens* by Ruiz-García et al. [9]. Morphological, physiological, chemotaxonomic, and phylogenetic interrelations have indicated that the two taxa are highly similar [2,7,10,11]. Furthermore, detailed examinations of the members of these two species, including phenotype analysis, phylogenetics, fatty acid methyl ester (FAME) analysis, DNA–DNA hybridization, microarray-based comparative genomic hybridization, genomic analysis, HPLC-electrospray ionization MS, and MALDI-TOF MS have revealed *B. velezensis* as plant-associated *B. amyloliquefaciens* subsp. *plantarum* subsp. nov., and *B. amyloliquefaciens* as non-plant-associated *B. amyloliquefaciens* subsp. *amyloliquefaciens* subsp. nov., respectively [12]. Subsequently, *B. methylotrophicus*, *B. amyloliquefaciens* subsp. *plantarum,* and *B. oryzicola* were reclassified later as heterotypic synonyms of *B. velezensis*, while *B. amyloliquefaciens* subsp. *amyloliquefaciens* was considered as *B. amyloliquefaciens* [13].

The earliest description differentiated *B. velezensis* and *B. amyloliquefaciens*, as well as other closely related taxa based on phenotypic and genetic differences [9], but in many cases, these taxonomical descriptions were later revised according to the current state of *Bacillus* taxonomy. A typical example for this is the identification history of strain DSM 23117^T^, which was first identified as *B. amyloliquefaciens* in 2008 [14], later revised as *B. amyloliquefaciens* subsp. *plantarum* in 2011 [12], and finally reclassified as *B. velezensis* in 2016 based on DNA–DNA hybridization, as well as phenotypic and phylogenetic analyses [13]. Furthermore, this statement was strongly confirmed for this strain using molecular methods [1]. Therefore, this well-defined strain was used as the type strain of *B. velezensis* in our study, although it is still named as *B. amyloliquefaciens* in several recent publications and GenBank sequences [15,16].

Over the last 15 years, interest in understanding the genetic relationship of the two taxa has led to many studies being published [1,3,10,11,12,13,14,17,18,19,20]. The two taxa share similar morphological, physiological, and phenotypic traits (Table 1) as well as 16S rRNA gene sequences, tetranucleotide frequency distribution, and DNA G+C contents [1,13]. Their average nucleotide identity and average amino acid identity is approximately 93.6–94.5% and 97.8% similarity, respectively [1,17], and their high DNA–DNA relatedness values showed 20 [9], 55 [1,13,17], or 80% similarity [14] in various studies. Strains *B. velezensis* DSM 23117^T^ and *B. amyloliquefaciens* DSM 7^T^ share 3345 genes in their core genomes, which have 97.89% similarity at the amino acid level [12]. Furthermore, the phylogenomic tree based on the core genome (799 genes) also indicates their close genetic relationship [13]. Both taxa are characterized by substantial production of secondary metabolites via non-ribosomal synthesis. However, only *B. velezensis* contains gene clusters synthesizing macrolactin and difficidin, which are lacking in *B. amyloliquefaciens* [1]. On the other hand, only *B. amyloliquefaciens* contains the *amyA* gene for industrial starch-liquefying α-amylase [12].

Previously, the whole-cell FAME profiles of *B. velezensis* and *B. amyloliquefaciens* have been studied only to a limited extent, therefore FAs were not considered as biomarkers distinguishing between them [2,10,11]. FAs are part of the bacterial cell membrane structure, and specific FAs and their ratios in cellular membranes have usually been revealed as biomarkers to distinguish closely related species of bacteria [2,4]. This study considered the possibility of using the cellular FAs with the application of a Sherlock chromatographic analysis system (CAS) as a taxonomic and diagnostic tool. The method, using FAs of 9–20 carbons in length and automated GC analysis, qualitatively and quantitatively analyzes bacterial whole-cell FAME [21]. Since the Sherlock CAS has developed, it has become capable of performing cost-effective, sensitive, reliable, and rapid analyses with a small amount of cell mass.

This study included DNA sequencing, as well as phylogenetic and FAME analyses with the aim of providing a complementary tool to distinguish *B. velezensis* and *B. amyloliquefaciens* based on their cellular FAs.

## 2. Materials and Methods

### 2.1. Bacillus Strains and Growth Conditions

*Bacillus* type strains including DSM 7^T^, DSM 1061^T^, and DSM 23117^T^ were obtained from the DSMZ (German Collection of Microorganisms and Cell Cultures, Braunschweig, Germany), while the *Bacillus* field strains were isolated according to Vágvölgyi et al. [22]. Briefly, soil samples (5 g) were collected from agricultural fields and suspended in 50 mL of 1% NaCl solution with intensive mixing with a glass rod, then the suspensions were allowed to pellet for a minute. The supernatants were used to make a dilution series. Fifty µL of each diluted sample was spread onto the surface of yeast extract–glucose (YEG) medium (glucose 0.2%, yeast extract 0.2%, bacto agar 2%) supplemented with 50 μg mL^−1^ nystatin to suppress fungi. After 7 days of incubation, the dominant bacterial colony morphotypes were picked and cleaned until homogeneity on YEG medium. The isolated strains were deposited in the Szeged Microbiology Collection (SZMC, http://szmc.hu, accessed on 29 December 2021) of the Department of Microbiology, University of Szeged, Hungary.

For molecular taxonomical investigations, *Bacillus* strains were cultured in YEG medium and incubated at 37 °C overnight. Before FA profiling, bacteria were inoculated on trypticase soy broth agar (TSBA, Becton, Dickinson and Company, Sparks, NV, USA) with the quadrant streaking method and incubated at 28 °C for 24 ± 2 h.

### 2.2. PCR Amplification of the gyrA and rpoB Genes

Total cellular DNA was extracted by the E.Z.N.A.^®^ Bacterial DNA Kit (Omega Bio-tek, Inc., Norcross, GA, USA) according to the manufacturer’s instructions. Amplification of the *gyrA* gene [22] was performed in 50 µL reaction mixtures containing 10 pmol of each primer (gyrAF: CAGTCAGGAAATGCGTACGTCCTT; gyrAR: CAAGGTAATGCTCCAGGCATTGCT), 10 nmol dNTP mix, 2 µL template DNA, 5 µL 10× PCR buffer, 6 µL of 25 mM MgCl_2_, and 1 U of Taq DNA polymerase (Thermo Fisher Scientific, Waltham, MA, USA). The PCR thermocycler (Doppio, VWR International GmbH, Darmstadt, Germany) was set to an initial denaturation step at 94 °C for 2 min, 30 cycles of denaturation at 94 °C for 30 s, annealing at 53 °C for 45 s and extension at 72 °C for 60 s, and a final extension at 72 °C for 5 min. The amplification of the *rpoB* gene [23] was conducted in 50 µL reaction mixtures containing 20 pmol of each primer (rpoBF: AGGTCAACTAGTTCAGTATGGACG; rpoBRO: GTCCTACATTGGCAAGATCGTATC), 10 nmol dNTP mix, 2 µL template DNA, 5 µL 10× PCR buffer, 6 µL of 25 mM MgCl_2_, and 0.4 U of Taq DNA polymerase. The PCR cycling parameters included an initial denaturation step at 94 °C for 2 min, 30 cycles of denaturation at 94 °C for 30 s, annealing at 57 °C for 30 s, extension at 72 °C for 50 s, and a final extension at 72 °C for 5 min. Sequencing of the amplified DNA fragments was performed on an ABI 3730XL sequencer (Thermo Fisher Scientific, Waltham, MA, USA) using Sanger sequencing.

### 2.3. Phylogenetic Analysis

Sequences were analyzed using the Mega X software [24]. The NCBI Nucleotide BLAST similarity search was carried out at https://blast.ncbi.nlm.nih.gov/Blast.cgi, accessed on 29 December 2021. Alignments were performed using the MUSCLE algorithm. The phylogenetic tree was inferred using the Neighbor-Joining method [24] with 1000 bootstrap replicates. The evolutionary distances were computed using the Tamura-Nei method and the units of the number of base substitutions per site [25]. The rate variation among sites was modeled with a gamma distribution (shape parameter = 0.5).

### 2.4. The Fatty Acid Methyl Ester (FAME) Analysis

The MIDI Sherlock^®^ Microbial Identification System (MIS, MIDI Inc., Newark, NJ, USA) was applied for the data acquisition [21]. The composition of whole-cell FAs was determined by the Sherlock CAS Software ver. 6.4 (Microbial ID Inc., Newark, DE, USA) operating through the LabSolution ver. 5.97 software on a Nexera GC-2030 gas chromatograph equipped with an AOC-20i Plus autoinjector (Shimadzu, Kyoto, Japan). For the separation of the FAs, the RTSBA6 method provided by the manufacturer was applied on a HP-Ultra 2, 25 m × 0.2 mm × 0.33 µm film thickness, fused, silica capillary column (Agilent, Santa Clara, CA, USA). Injector and detector temperatures were 250 °C and 300 °C, respectively. Carrier gas was hydrogen at a flow rate of 1.48 mL min^−1^, while the detector gases were nitrogen (make up), oxygen and hydrogen with the follow flows of 30, 30, and 350 mL min^−1^, respectively. Samples were introduced in an injection volume of 2 µL in split mode with a 40:1 split ratio. The oven program started at 168.1 °C, which ramped up to 291 °C with 28 °C min^−1^, and then up to 300 °C with 60 °C min^−1^, holding at this temperature for 1.50 min. The total column oven program time was 6.04 min. The 1300-C rapid calibration standard mix (Microbial ID Inc., Newark, DE, USA) was used for retention time calibration and system suitability purposes. The *B. subtilis* strain ATCC 6633^T^ and pure hexane were considered as the quality control and the negative control, respectively. Whole-cell FAME profiles were analyzed by the library RTSBA6.21 (Microbial ID Inc., Newark, DE, USA).

### 2.5. Sample Pretreatment

Sample processing was carried out according to the Sherlock^TM^ Operating CAS Manual [21]. Briefly, 20–40 mg of cells was harvested and placed in a clean glass tube. Then, 1 mL of reagent 1 (45 g NaOH, 150 mL of methanol and 150 mL of distilled water) was added to the sample and heated at 95–100 °C in a water bath (Precision water bath NB-301, HandyLAB^®^System, N-BIOTEK, Bucheon-si, Korea). After 5 min, the sample was removed from the water bath, vortexed and heated for an additional 25 min. The sample was mixed with 2 mL of reagent 2 (325 mL of 6.0 N HCl, 275 mL of methanol) and incubated at 80 °C in a water bath for 10 min. Subsequently, 1.25 mL of reagent 3 (200 mL of hexane, 200 mL of methyl tert-butyl ether) was added and the derivatized FAs were extracted for 10 min in a laboratory rotator (Rotator drive STR4 Stuart, Cole-Parmer^TM^, Vernon Hills, IL, USA). The organic (upper) phase was recovered and washed with 3 mL of reagent 4 (10.8 g NaOH, 900 mL distilled water) for 5 min in a laboratory rotator. The resulting organic (upper) phase from the tube was transferred to a clean vial for GC analysis.

### 2.6. Statistical Analysis

The library generation function of MIS Sherlock Command Centre ver. 6.4 was applied to install a new library of *Bacillus* named RTSBA7. New entries of *Bacillus* species were added by a statistical summary of a set of related samples. The MIS Sherlock Command Centre had been applied also for data analysis. The Dendrogram cluster analysis technique, using Euclidian distance (ED) metric, was applied for determining the distance between individual FAs, producing unweighted pair matchings based on FA compositions. The results were displayed graphically to depict the relatedness between organisms. The 2D plot cluster analysis technique using a principal component (PC) analysis was used to separate groups of samples in an n-dimensional space.

## 3. Results

### 3.1. The Classification of B. velezensis and B. amyloliquefaciens Based on Molecular Markers

The application of the type strains is necessary for the development of a reliable FA profiling method capable of differentiating between the closely related *B. velezensis* and *B. amyloliquefaciens* species, and the use of a confirmatory method is also essential for the classification of unknown *Bacillus* isolates. Therefore, as a confirmatory analysis, the partial sequences of the genes encoding the subunit A protein of DNA gyrase (*gyrA*) and the RNA polymerase beta-subunit (*rpoB*) were determined to identify the isolated strains. The BLASTN comparison showed high similarities between the examined strains (Table 2) and corresponding records of *B. velezensis* and *B. amyloliquefaciens* strains in the GenBank database. Accordingly, the *gyrA* and *rpoB* sequences of *B. velezensis* DSM 23117^T^ and the isolated strains displayed approximately 100% similarity with both *B. velezensis* and *B. amyloliquefaciens* records in GenBank. Furthermore, the *gyrA* and *rpoB* genes of *B. amyloliquefaciens* DSM 7^T^ and DSM 1061^T^ strains shared also approximately 100% similarities with numerous *B. amyloliquefaciens* and *B. velezensis* strains in GenBank, respectively. Thus, the classification using both *gyrA* and *rpoB* genes revealed high relatedness values between *B. velezensis* and *B. amyloliquefaciens*, and the BLAST results also showed the presence of sequences from possibly misidentified strains deposited in the GenBank.

To investigate the relatedness of strains, a phylogenetic tree was also built using the Neighbor-Joining method with 1000 bootstrap replicates. The *B. subtilis* ATCC 6633 strain was considered as the outgroup. Analyses with the *gyrA* sequences (Appendix A), *rpoB* sequences (Appendix A), and their concatenation (Figure 1) generated similar phylogenetic trees without notable differences. The analysis separated *B. velezensis* and *B. amyloliquefaciens* strains into two corresponding clades of the phylogenetic tree in the case of the type strains; other databases collected strains and the field isolates (Appendix A) with the bootstrap values of 83 and 98%, respectively (Figure 1). The phylogenetic tree, based on *gyrA* and *rpoB* sequences, shows the existence of two tightly related monophyletic groups: (1) *B. velezensis,* containing our field *Bacillus* isolates, type strain DSM 23117^T^ together with the other reference strains At1, AS43.3, BIM B-439D, AP183, KKLW, SQR9, S141, QST713, BvL03, LF01, WRN014, and SGAir0473; (2) *B. amyloliquefaciens,* containing type strain DSM 7^T^ and DSM 1061^T^ together with the other reference strains LL3, TA208, and XH7.

### 3.2. FAME Profiles of B. velezensis and B. amyloliquefaciens Strains

The content of FAs was revealed (Table 3, Appendix A) and the features were constructed from the MIS analysis of sixteen *B. velezensis* strains (n = 3) and two *B. amyloliquefaciens* strains (n = 25). The 15:0 iso (13-methyltetradecanoic), 15:0 anteiso (12-methyltetradecanoic), 16:0 (n-hexadecanoic), 17:0 iso (15-methylhexadecanoic), and 17:0 anteiso (14-methylhexadecanoic) have been primary FA components in both taxa. The predominant content of 15:0 iso and 15:0 anteiso are 30.39 ± 2.53 and 32.13 ± 2.33 (%) in *B. velezensis* and 27.85 ± 1.67 and 31.88 ± 1.98 (%) in *B. amyloliquefaciens*, respectively. Besides, the minor content of FA 14:0 iso (12-methyltridecanoic), 14:0 (n-tetradecanoic), 16:0 iso (14-methylpentadecanoic), 16:1 ω11c (cis-5-hexadecenoic), and 17:1 iso ω10c ((6Z)-15-methyl-6-hexadecenoic) are approximately from 1.0 to 3.5% in both taxa (Table 3). Especially, the FA 16:0, 17:0 iso, and 17:0 have drawn a distinction between *B. velezensis* and *B. amyloliquefaciens*. In the case of *B. velezensis*, the proportions of 16:0, 17:0 iso, and 17:0 anteiso were 12.53 ± 1.82, 8.52 ± 0.96, and 5.50 ± 0.85 (%), respectively, which were compared with those for the *B. amyloliquefaciens* strains, which were 4.55 ± 0.54, 15.98 ± 1.95, and 8.97 ± 0.73 (%), respectively.

### 3.3. Differentiation of FAME Profiles between B. velezensis and B. amyloliquefaciens

The FA profiles were consistently typical and distinguishable between *B. velezensis* and *B. amyloliquefaciens*. Principal component analysis enabled us to look at data with high dimensionality and observe the most critical aspects of the data in two or three dimensions. The 2D plot built from PC 1 and PC 2 (Figure 2) showed a separation of the two taxa in n-dimensional space. The group A represented FA components of *B. amyloliquefaciens* with ED^2^ (Euclidian distance squared) ~ 36, and the group B represented FAs of *B. velezensis* with ED^2^ ~ 100. A group with calculated ED^2^ ≤ 100 was considered as the same species. The FA profiles of the two taxa could be distinguished into two separate groups of strains.

Additionally, according to the criterion established by MIS, when the similarity index (SI) is larger than 0.5 and separated from other organisms from the library by at least 0.100, the sample is considered identified. The SI value generated from calculations of distance in multi-dimensional space illustrated the relation between analyzing FA profiles and the mean FAs of library’s database as its match. In our case, all identified samples exhibited high matches with SI > 0.5 and well-separated SI (>0.1) confirming that the method is reliable with high confidence.

The dendrogram analysis drawn from diverse FA profiles of *Bacillus* species showed a relationship between them (Figure 3) via the ED index. These profiles were obtained from the Sherlock library that had been built from diverse strains (more than 20 strains) within each species. The samples were collected from across the world to avoid potential geographic bias and carefully analyzed with many replications to make the entries of the library [21]. The FA profiles of *B. velezensis* and *B. amyloliquefaciens*—together with *B. agaradhaerens*, *B. pumilus*, *B. licheniformis,* and *B. subtilis*—formed one cluster, which could be distinguished from other clusters. Furthermore, FAs of *B. velezensis*, *B. agaradhaerens,* and *B. amyloliquefaciens* formed a tight phylogenetic branch, which showed the highest phenotypic similarity. The FA profiles of group-related species were highly similar and were also determined previously [26,27,28].

## 4. Discussion

The classification of *B. velezensis* and *B. amyloliquefaciens* has usually been a particularly confounding taxonomic problem. Moreover, it was also concluded in previous reports that the whole-cell FAME profiles had not yielded satisfying results for discriminating these two species [2,10,11]. However, our research efforts, aimed at developing a whole-cell FAME profile-based method for distinguishing both taxa, led to other conclusions.

The *gyrA* and *rpoB* sequences proved to be effective for resolving these closely related species of the *B. subtilis* group [1,19,26]. The previous use of *gyrA* [12,19] and *rpoB* [1,11] as phylogenetic markers had drawn clear distinction between the two taxa. Accordingly, their highly conserved *rpoB* sequences shared approximately 98% similarity [1], but the NJ tree obtained from their *gyrA* sequences distinguished them with bootstrap values of 100 and 52% [12].

In the present study, *gyrA* and *rpoB* sequences were amplified and aligned to determine *Bacillus* strains to the species level. The result of BLAST alignments showed high relatedness between the sequences of the studied strains and records of both *B. velezensis* and *B. amyloliquefaciens* from the GenBank database. Thus, the discrimination of the two taxa were confused, because, unfortunately, several *B. velezensis* strains are still named as *B. amyloliquefaciens* in the GenBank and vice versa, which makes the identifications difficult. Therefore, for the detailed analysis, the partial sequences of *gyrA* and *rpoB* genes were concatenated and included in the phylogenetic analysis for comparative purposes. The selection of reference sequences was carefully considered according to previous classifications to avoid misinterpretations [2,7,12,16,29]. The studied strains clustered into two separate clades on the phylogenetic tree (Figure 1) differentiating *B. velezensis* strains from the cluster of strains related to *B. amyloliquefaciens* DSM 7^T^ and DSM 1061^T^. Then the result of this classification was used for developing the identification method based on FA profiles.

In the current work, FAs were identified based on their estimated carbon lengths determined relative to the calibration standard and by comparing with the peak table. It could be concluded that certain deviations can be found between our results and previously published FA profiles [2,10,11]. In general, both taxa possess a higher content of branched-odd FAs, including 15:0 iso, 15:0 anteiso, 17:0 iso, and 17:0 anteiso than other FAs. The presence of branched-chain FAs is expected to increase the membrane’s fluidity because of their low melting point temperatures, and are already remarkable biomarkers used in *Bacillus* taxonomy [5]. The 15:0 iso and 15:0 anteiso FAs have shared a prominent proportion, similar to other species within the “*B. subtilis* species complex” [9] and their high ratio has been indicated as a common feature in *Bacillus* species [5]. These FAs had been considered as being distinguishable features between many other *Bacillus* species reported previously [25]. Discriminating biomarkers useful for distinguishing between the two taxa were 14:0, 16:0, 16:0 iso, 17:0 iso, and 17:0 anteiso. The FA profiles of *B. velezensis* could be characterized by higher 14:0 and 16:0 contents and lower 16:0 iso, 17:0 iso and 17:0 anteiso contents in comparison to *B. amyloliquefaciens* (Figure 4).

Our comprehensive study proved that these features are valuable taxonomical biomarkers with high discriminatory power, even though previous studies reported on the insufficiency of FA components in the discrimination of the two taxa. In the previous studies, certain involved isolates were misidentified and the novel classification of these strains, proven by many recent scientific contributions, helped us in giving a better conclusion for this issue. As shown in Table 4, the present investigation was similar to the report of Wang et al. [14]. However, it was a misapprehension that the publication considered *Bacillus* strain BCRC 14193 as *B. amyloliquefaciens*, which was later reclassified as *B. velezensis* by Dunlap et al. [13]. Currently, considering strain BCRC 14193 as *B. velezensis*, a distinguishable FA comparison was obviously drawn between the two taxa (Table 4). In 2011, a comparison among FAs from six strains of *B. velezensis* and five strains of *B. amyloliquefaciens* had been reported, with varied cellular FA compositions showing differences between the two taxa in the case of FA 14:0, 16:0, and 16:0 iso [12]. However, *B. velezensis* DSM 23117^T^ and *B. amyloliquefaciens* DSM 7^T^ contained a high content of FA 17:0, but lacked 17:0 anteiso [12], which made a difference with other *Bacillus* strains, and it is difficult to find a comparison, as well as the relatedness with our present study, due to insufficient data. The FA profiles in our study also shared high similarity to those of *B. velezensis* sp. nov. CR-502^T^ and *B. amyloliquefaciens* DSM 7^T^ with some minor differences [9]. It is interesting that only *B. amyloliquefaciens* DSM 7^T^, from the publication of Ruiz-García et al. [9], contained FA 16:1 ω5c, 16:1 ω9c, and 17:1 iso ω7c, which had not been detected by other authors.

A high-quality library plays an important role in the classification. This study carefully constructed the library RTSBA7 from 16 strains of *B. velezensis* (n = 3) and 2 strains of *B. amyloliquefaciens* (n = 25) and from the data available in the RTSBA6, containing *B. amyloliquefaciens* using various strains constructed by MIDI (Table 2).

Altogether a total of 31 FAs were detected in *B. velezensis* (Appendix A) and 38 FAs were determined in *B. amyloliquefaciens* (Appendix A) during the analyses. The calculations were interfered with by occasionally detected peaks irregularly observed on the chromatograms with small peak areas; however the MIS analysis, with many replicates of samples, can detect and remove these variations, creating precise whole-cell FA features (Figure 4). For example, in the case of *B. velezensis*, the MIS analysis did not consider FA 9:0, because there were only 3 out of 48 samples containing it, with a low mean (=0.06) and high SD/mean (=4.15). Otherwise, FA 14:0 (2.87 ± 0.70 (%)) detected from 1.69 to 4.89% in all 48 samples was considered as a valuable parameter. Accordingly, characterizations of both taxa have contained 13 FAs as features of the analyses (Table 2), which are reported in the Appendix A.

Once a sample has been analyzed by Sherlock, its FA composition can be matched with those of known organisms that are stored in the library. The Sherlock Library search lists the most likely matches to the query composition, and provides an SI for each match, which is a numerical value expressing how closely the FA composition of the query compares with the mean FA composition of the strains used to create the library entry listed as its match. The database search presents the best matches and associated SI. This value is a software-generated calculation of the distance in multi-dimensional space between the profile of the query and the mean profile of the closest library entry. Our results showed good matches between the experimental samples and the FA composition in the library, with SI > 0.5 and well-separated SI (>0.1). In addition, the FA compositions could be separated between the two taxa and among other *Bacillus* species. 

## 5. Conclusions

In our study, a method using FAs of 9–20 carbons in length and automated GC analysis were developed to qualitatively and quantitatively analyze the bacterial whole-cell FAs as taxonomical markers. To the best of our knowledge, this is the first time that the method based on whole-cell FA profiles operated by MIS has been applied to distinguish between *B. velezensis* and *B. amyloliquefaciens* with comprehensive evidence. By taking advantage of the current knowledge regarding biomarkers, the FA-based identification proved to be applicable for the differentiation between these closely related species. Our experiments provided a cost-effective, reliable, and fast-automated solution for discrimination between these taxa.

## Figures and Tables

**Figure 1 microorganisms-10-00418-f001:**
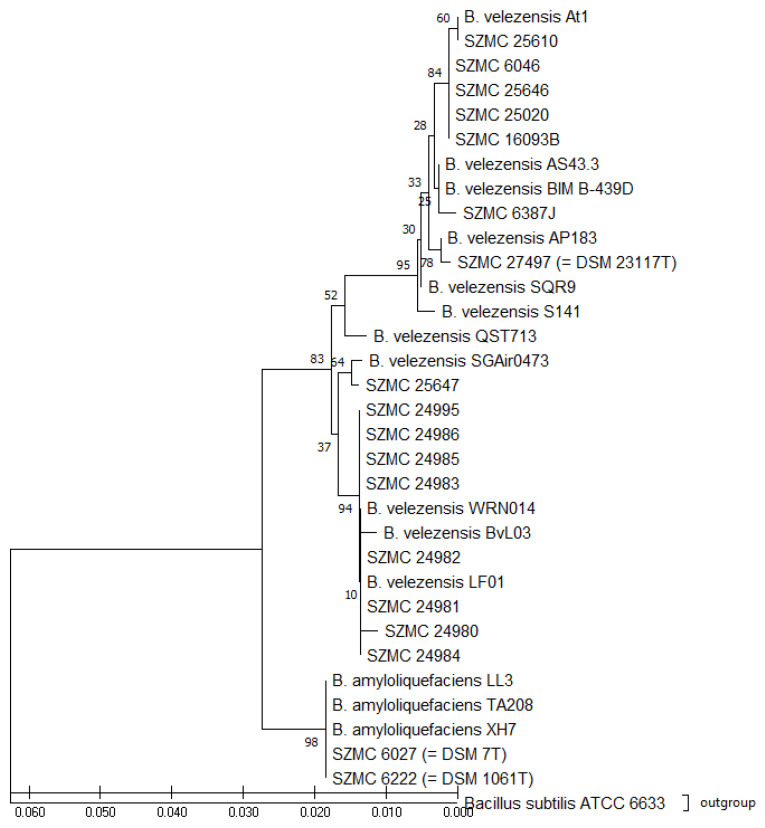
Neighbor-Joining phylogenetic tree based on the concatenation of *gyrA* and *rpoB* gene sequences. Evolutionary distances were computed by the Tamura-Nei method. Bars, 0.010 substitutions per nucleotide position.

**Figure 2 microorganisms-10-00418-f002:**
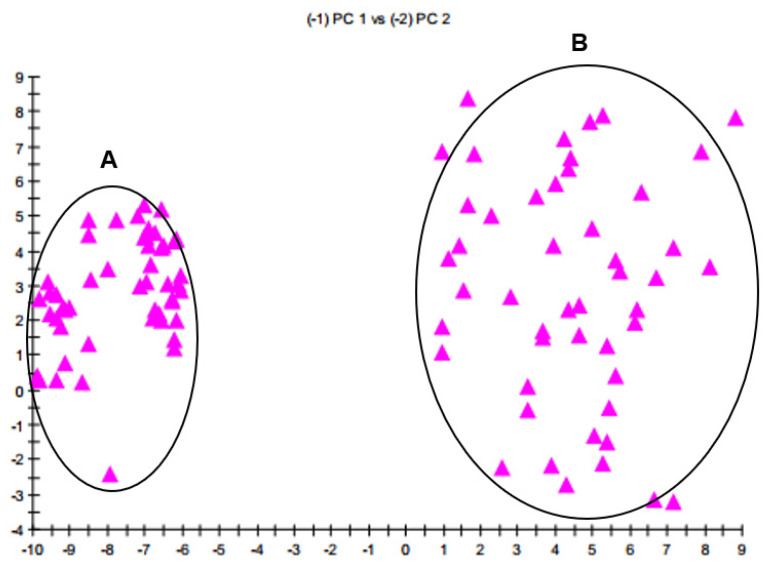
The 2D plot between FA components of *B. velezensis* and *B. amyloliquefaciens*. (**A**) FA components from the group of *B. amyloliquefaciens*, (**B**) FA components from the group of *B. velezensis*.

**Figure 3 microorganisms-10-00418-f003:**
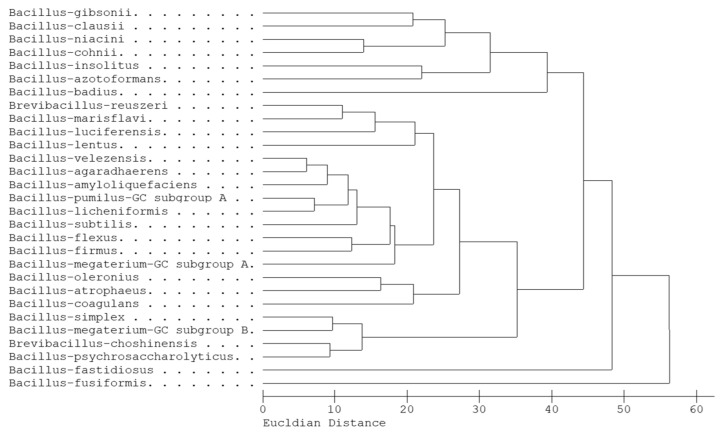
The relationship of FA profiles among the *Bacillus* species in the MIS library.

**Figure 4 microorganisms-10-00418-f004:**
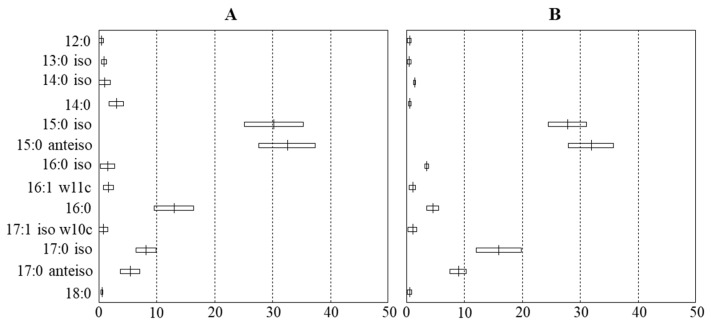
Comparison charts of (**A**) *B. velezensis* and (**B**) *B. amiloliqefaciens* based on FA profiles created in the MIS library.

**Table 1 microorganisms-10-00418-t001:** Characteristics of *B. velezensis* and *B. amyloliquefaciens* determined by different techniques.

Characteristics ^a^	*B. velezensis* ^b^	*B. amyloliquefaciens* ^b^	References
Pigmentation	Creamy white	Creamy white	[9]
Oxidase	+	+	
Acid in API system from:			
- Glycogen	+	nd	
- Lactose	+	+	
- Melibiose	−	+	
- Methyl α-ᴅ-glycoside	+	+	
- ᴅ-Raffinose	+	+	
- ᴅ-Turanose	−	+	
Hydrolysis of			
- Tween 20	−	+	
- Tween 80	−	nd	
- DNA	−	−	
Arginine dihydrolase	−	−	
ONPG	+	−	
Non-ribosomally synthesized secondary metabolites			[1,10,12]
- Surfactin	+	+	
- Macrolactin	+	−	
- Bacillaene	+	+	
- Fengycin	+	−	
- Difficidin	+	−	
- Bacillibactin	+	+	
- Bacilysin	+	+	
Ribosomally synthesized antimicrobial compounds			
- Sublancin	−	−	
- Subtilosin	−	−	
- Amylocyclicin	+	+	
- Plantazolicin	+	−	
Plant colonization	+	−	[12]
AZCL-HE cellulose liquefaction	+	−	
Growth in lactose minimal medium	+	+	
Amylase AmyA	−	+	
Amylase AmyE	+	−	
Cellulase BglC	+	−	
Xylanase XynA	+	−	
16S rRNA gene sequence similarity (%)	99.7%		[14]
*rpoB* gene sequence similarity (%)	>98%		[1]
*gyrB* gene sequence similarity (%)	>95.5%		[14]
DNA relatedness value (%)	74–84%		[14]

^a^ ONPG: *O*-nitrophenyl β-ᴅ-galactopyranoside, AZCL-HE: endocellulase activity determined by using insoluble azurine cross-linked (AZCL)-HE-cellulose. ^b^ +: detected, −: not detected, nd: not determined.

**Table 2 microorganisms-10-00418-t002:** *Bacillus* strains used in the FA profiling study.

Strains ^a^	Genbank Accession Number		Origin
	*gyrA*	*rpoB*	
*B. velezensis*			
SZMC 24980	OK256097	OK256115	soil sample from pepper field, Totovo selo, Serbia
SZMC 24981	OK256098	OK256116	soil sample from pepper field, Totovo selo, Serbia
SZMC 24982	OK256099	OK256117	soil sample from pepper field, Totovo selo, Serbia
SZMC 24983	OK256100	OK256118	soil sample from pepper field, Totovo selo, Serbia
SZMC 24984	OK256101	OK256119	soil sample from pepper field, Cantavir, Serbia
SZMC 24985	OK256102	OK256120	soil sample from pepper field, Cantavir, Serbia
SZMC 24986	OK256103	OK256121	soil sample from tomato field, Cantavir, Serbia
SZMC 24995	OK256104	OK256122	soil sample from tomato field, Cantavir, Serbia
SZMC 25020	OK256105	OK256123	soil sample from tomato field, Cenej, Serbia
SZMC 25646	OK256106	OK256124	pea rhizosphere, Madaras, Hungary
SZMC 25647	OK256107	OK256125	pea rhizosphere, Madaras, Hungary
SZMC 25610	OK256108	OK256126	maize rhizosphere, Vaszar, Hungary
SZMC 6046	OK256109	OK256127	tomato rhizosphere, Hungary
SZMC 16093B	OK256110	OK256128	tomato rhizosphere, Hungary
SZMC 6387J	OK256111	OK256129	tomato rhizosphere, Hungary
DSM 23117^T^ (=BGSC 10A6 = FZB42 = LMG 26770 = SZMC 27497)	OK256112	OK256130	plant pathogen-infested soil of a sugar beet field, Brandenburg, Germany
*B. amyloliquefaciens*			
DSM 7^T^ (=ATCC 23350 = SZMC 6027)	OK256113	OK256131	soil and industrial amylase fermentations, Japan
DSM 1061^T^ (=IAM 1523 = SZMC 6222)	OK256114	OK256132	unknown origin
*B. subtilis*			
ATCC 6633^T^	CP039755.1	CP039755.1	Japan

^a^ ATCC—American Type Culture Collection, BGSC—Bacillus Genetic Stock Center, DSM—German Collection of Microorganisms and Cell Cultures, FZB—Research Center Borstel, IAM—Institute of Applied Microbiology, University of Tokyo, LMG—Belgian Coordinated Collections of Microorganisms/LMG Bacteria Collection, SZMC—Szeged Microbiology Collection.

**Table 3 microorganisms-10-00418-t003:** Cellular fatty acid compositions (mean (%) ± SD).

Feature/FA	*B. velezensis*	*B. amyloliquefaciens*
12:0	0.48 ± 0.23	0.54 ± 0.17
13:0 iso	0.89 ± 0.22	0.50 ± 0.19
14:0 iso	1.18 ± 0.58	1.44 ± 0.11
14:0	2.87 ± 0.70	0.61 ± 0.14
15:0 iso	30.39 ± 2.53	27.84 ± 1.65
15:0 anteiso	32.13 ± 2.33	31.92 ± 1.98
16:0 iso	1.70 ± 0.77	3.51 ± 0.19
16:1 ω11c	1.65 ± 0.42	1.09 ± 0.28
16:0	12.53 ± 1.82	4.57 ± 0.55
17:1 iso ω10c	0.85 ± 0.47	1.07 ± 0.37
17:0 iso	8.52 ± 0.96	15.92 ± 1.96
17:0 anteiso	5.50 ± 0.85	8.99 ± 0.73
18:0	0.60 ± 0.14	0.59 ± 0.23

**Table 4 microorganisms-10-00418-t004:** Comparison of FA components of *B. velezensis* and *B. amyloliquefaciens* reported in the literature.

Feature	This Study		Ruiz-García et al. [9]		Wang et al. [14]				Borriss et al. [12]	
	*B. v.* ^a^	*B. a.* ^b^	*B. v.*	*B. a.*	*B. v.*	*B. v.*	*B. a.*	*B. a.*	*B. v.*	*B. a.*
			CR-502^T^	DSM 7^T^	BCRC 17467^T^	BCRC 14193	BCRC 11601^T^	BCRC 17038	DSM 23117^T^	DSM 7^T^
12:0	0.48	0.54	-	-	-	-	-	-	-	-
13:0 iso	0.89	0.50	0.87	-	-	-	-	-	0.31	0.38
14:0 iso	1.18	1.44	1.08	2.46	-	1.3	1.5	1.7	0.43	0.99
14:0	2.87	0.61	2.96	-	3.8	3.1	-	-	1.21	0.36
15:0 iso	30.39	27.84	29.86	30.50	30.4	24.0	26.3	23.7	31.00	40.29
15:0 anteiso	32.13	31.92	32.70	36.48	27.6	28.7	32.3	33.8	31.73	28.32
16:0 iso	1.70	3.51	1.31	4.52	1.0	2.2	3.8	4.3	1.01	2.13
16:1 ω5c	-	-	-	2.14	-	-	-	-	-	-
16:1 ω7c	-	-	-	-	-	-	-	-	0.19	0.42
16:1 ω9c	-	-	-	0.62	-	-	-	-	-	-
16:1 ω11c	1.65	1.09	4.42	-	3.5	2.7	1.7	-	2.59	1.23
16:0	12.53	4.57	13.41	4.52	18.3	19.0	5.8	7.0	7.60	3.02
17:1 iso ω7c	-	-	-	1.67	-	-	-	-	-	-
17:1 iso ω10c	0.85	1.07	1.44	-	1.3	1.3	1.7	-	2.70	2.59
17:0 iso	8.52	15.92	7.67	9.01	7.8	10.3	16.3	17.6	12.11	13.14
17:0 anteiso	5.50	8.99	4.27	7.06	3.4	5.4	9.0	10.0	-	-
17:0	0.17	0.22	-	-	-	-	-	-	7.70	6.46
18:0	0.60	0.59	-	-	-	1.1	-	-	-	-

^a^*B. v.*: *B. velezensis*; ^b^*B. a.*: *B. amyloliquefaciens*.

## Data Availability

DNA sequence data supporting the reported results can be found in the NCBI GenBank database (https://www.ncbi.nlm.nih.gov/genbank, 29 December 2021) under the accession numbers listed in Table 1.

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
