# Peer review of "Discrimination between the Two Closely Related Species of the Operational Group B. amyloliquefaciens Based on Whole-Cell Fatty Acid Profiling"

_microorganisms, 2022, doi:10.3390/microorganisms10020418_

Round 1

Reviewer 1 Report

Reviewer 1:

Comments to the Author

General comments: The manuscript by Huynh et al., (submitted) discriminates between two closely related species of the B. amyloliquefaciens group based on whole cell fatty acid profiling. This is an interesting study which proposes a fast and easy automated identification tool based on cellular FA profiles to distinguish between B. velezensis and B. amyloliquefaciens. For this manuscript to be accepted, kindly address specific comments below.

Specific comments

Lines 1-4: Consider swapping words in the title to read: “Discrimination Between the Two Closely Related Species of B. amyloliquefaciens Operational Group Based on Whole-Cell Fatty Acid Profiling”

Lines 42-44: Consider fragmenting this sentence; it is ambiguous as it is now. The first opening sentence should stop after “…the operational group B. amyloliquefaciens.” The rest should form another sentence.

Line 45: delete “been”

Line 46: what “products” are being referred to here? Biocontrol? biostimulation? Better to be specific

Line 52: replace “firstly” with “first”

Line 53: replace “firstly” with “first”

Line 57: delete “the” after “Furthermore”

Line 63: replace “After that” with “Subsequently”

Line 70: replace “identified firstly” with “first identified”

Line 95: replace “been usually” with “usually been”

Line 98: Rephrase: “…the extension of the Sherlock….”

Line 100: replace “As” with “Since”

Line 101: This is a disturbing phrase “it has possibly become”. It is better to say “it has become” OR “it is fast becoming” otherwise, the basis of your study becomes questionable.

Line 113: replace “let” with “allowed”

Line 142: replace “with” with “using”

Line 144: Delete “The” at the start of the sentence.

Line 145: delete “for”

Line 146: replace “by” with “using”

Lines 162-163: Rephrase: “The injection volume (2 μl), was introduced ….

Line 169: Delete “The” at the start of the sentence.

Line 172: Delete “The” at the start of the sentence.

Line 195: insert “a” before “n-dimensional space”.

Line 217: replace “as being an” with “as the”

Line 220: This phrase needs review because it is unclear “…into two corresponding separate of the phylogenetic tree,…”

Line 260: replace “allowed” with “enabled”

Line 261: replace “see” with “observe”

Lines 265-266:  Rephrase this, it is grammatically incorrect “aimed that a group is belonged in a same species”

Line 281: replace “over” with “across”

Line 297: replace “lead” with “led”

Line 299: Rephrase: “…had already been shown to be …”

Line 327: replace “similarly” with “similar”

Line 350: rephrase “…However, in that study, B. velezensis DSM 23117T and B. amyloliquefaciens DSM 7T contained…….”

Line 354: replace “the few data available” with “insufficient data”

Line 354: rephrase “shared also” with “also shared”

Supplementary material

Correct table title: “….data used in phylogeny construction”

Figure S1 and Figure S2: Remove “The” before “Neighbour-joining”

Author Response

Answers to Reviewer 1:

The authors wish to thank the Reviewer for the valuable comments and suggestion, which proved very helpful in improving our manuscript and preparing its revised version. The comments of the Reviewer were addressed as follows:

Reviewer: Lines 1-4: Consider swapping words in the title to read: “Discrimination Between the Two Closely Related Species of B. amyloliquefaciens Operational Group Based on Whole-Cell Fatty Acid Profiling”

Answer: The “operational group B. amyloliquefaciens” is a group name, which was proposed and used by previous studies. Therefore, we intend to keep this form to follow easier the history of this topic.

Reviewer:

Lines 42-44: Consider fragmenting this sentence; it is ambiguous as it is now. The first opening sentence should stop after “…the operational group B. amyloliquefaciens.” The rest should form another sentence.

Line 45: delete “been”

Line 46: what “products” are being referred to here? Biocontrol? biostimulation? Better to be specific

Line 52: replace “firstly” with “first”

Line 53: replace “firstly” with “first”

Line 57: delete “the” after “Furthermore”

Line 63: replace “After that” with “Subsequently”

Line 70: replace “identified firstly” with “first identified”

Line 95: replace “been usually” with “usually been”

Line 98: Rephrase: “…the extension of the Sherlock….”

Line 100: replace “As” with “Since”

Line 101: This is a disturbing phrase “it has possibly become”. It is better to say “it has become” OR “it is fast becoming” otherwise, the basis of your study becomes questionable.

Line 113: replace “let” with “allowed”

Line 142: replace “with” with “using”

Line 144: Delete “The” at the start of the sentence.

Line 145: delete “for”

Line 146: replace “by” with “using”

Lines 162-163: Rephrase: “The injection volume (2 μl), was introduced ….

Line 169: Delete “The” at the start of the sentence.

Line 172: Delete “The” at the start of the sentence.

Line 195: insert “a” before “n-dimensional space”.

Line 217: replace “as being an” with “as the”

Line 220: This phrase needs review because it is unclear “…into two corresponding separate of the phylogenetic tree,…”

Line 260: replace “allowed” with “enabled”

Line 261: replace “see” with “observe”

Lines 265-266:  Rephrase this, it is grammatically incorrect “aimed that a group is belonged in a same species”

Line 281: replace “over” with “across”

Line 297: replace “lead” with “led”

Line 299: Rephrase: “…had already been shown to be …”

Line 327: replace “similarly” with “similar”

Line 350: rephrase “…However, in that study, B. velezensis DSM 23117T and B. amyloliquefaciens DSM 7T contained…….”

Line 354: replace “the few data available” with “insufficient data”

Line 354: rephrase “shared also” with “also shared”

Supplementary material:

Correct table title: “….data used in phylogeny construction”

Figure S1 and Figure S2: Remove “The” before “Neighbour-joining”

Answer: All changes have been performed according to the Reviewer’s suggestions mentioned above.

Reviewer 2 Report

  1. Do rephrases lines 22-24
  2.  Overall the abstract and the manuscript need extensive English grammar modifications.
  3.  The introduction must be more structured with clear demarcation between similarities and dissimilarities, if needed subheadings can be used as well. 
  4. A small table can be introduced to highlight the similarities
  5. Although rare but if there are any microbiology/biochemical dissimilarity, that must also be enlisted here.

Author Response

Answers to Reviewer 2:

The authors wish to thank the Reviewer for the valuable comments and suggestion, which proved very helpful in improving our manuscript and preparing its revised version. The comments of the Reviewer were addressed as follows:

Reviewer: Do rephrases lines 22-24

Answer: The text has been rephrased as „Bacillus velezensis and Bacillus amyloliquefaciens are closely related members of the “operational group B. amyloliquefaciens”, a taxonomical unit above species level within the ”Bacillus subtilis species complex”.

Reviewer: Overall the abstract and the manuscript need extensive English grammar modifications.

Answer: The abstract and the manuscript were thoroughly checked, and English modifications were performed.

Reviewer:

The introduction must be more structured with clear demarcation between similarities and dissimilarities, if needed subheadings can be used as well.

A small table can be introduced to highlight the similarities.

Although rare but if there are any microbiology/biochemical dissimilarity, that must also be enlisted here.

Answer: To provide clear demarcation between similarities and dissimilarities of the two species, Table 1. entitled „Characteristics of B. velezensis and B. amyloliquefaciens determined by different techniques” has been introduced in the revised version of the manuscript. The new table also enlists microbiological and biochemical dissimilarities.